# Burden of Mesothelioma Deaths by National Income Category: Current Status and Future Implications

**DOI:** 10.3390/ijerph17186900

**Published:** 2020-09-21

**Authors:** Odgerel Chimed-Ochir, Diana Arachi, Tim Driscoll, Ro-Ting Lin, Jukka Takala, Ken Takahashi

**Affiliations:** 1Department of Environmental Epidemiology, Institute of Industrial Ecological Sciences, University of Occupational and Environmental Health, Kitakyushu City 807-8555, Fukuoka Prefecture, Japan; odgerel@med.uoeh-u.ac.jp; 2Asbestos Diseases Research Institute, Gate 3 Hospital Road, Concord, NSW 2139, Australia; diana.arachi@adri.org.au; 3Epidemiology and Occupational Medicine, School of Public Health, University of Sydney, Sydney, NSW 2006, Australia; tim.driscoll@sydney.edu.au; 4Department of Occupational Safety and Health, College of Public Health, China Medical University, Taichung 406, Taiwan; roting@mail.cmu.edu.tw; 5International Commission on Occupational Health, ICOH/INAIL Research Area, Via Fontana Candida, 1-00078 Monteporzio Catone (Rome), Italy; ICOHPresident@inail.it; 6University of Occupational and Environmental Health, Kitakyushu City 807-8555, Fukuoka Prefecture, Japan

**Keywords:** mesothelioma, asbestos, deaths, burden of disease, global health, estimation, developing countries, national income

## Abstract

**Background:** This study compares estimates of the global-level mesothelioma burden with a focus on how existing national mortality data were utilized and further assesses the interrelationship of country-level mesothelioma burden and asbestos use with national income status. **Methods:** Country-level mesothelioma deaths in the WHO Mortality Database as of December 2019 were analyzed by national income category of countries in terms of data availability and reliability. Numbers of mesothelioma deaths from the study of Odgerel et al. were reanalyzed to assess country-level mesothelioma death burdens by national income status. **Results:** Among 80 high-income countries, 54 (68%) reported mesothelioma to the WHO and 26 (32%) did not, and among 60 upper middle-income countries, the respective numbers (proportions) were 39 (65%) countries and 21 (35%) countries, respectively. In contrast, among 78 low- and lower middle-income countries, only 11 (14%) reported mesothelioma deaths while 67 (86%) did not. Of the mesothelioma deaths, 29,854 (78%) were attributed to high- and upper middle-income countries, and 8534 (22%) were attributed to low- and lower middle- income countries. **Conclusions:** The global mesothelioma burden, based on reported numbers, is currently shouldered predominantly by high-income countries; however, mesothelioma burdens will likely manifest soon in upper middle-income and eventually in low and lower middle-income countries.

## 1. Introduction

Asbestos-related diseases (ARD) have been characterized as a “global health” issue [1] and more recently a “global disaster” [2]. Although the situations of individual countries are mixed, the grouping of countries into *industrialized* and *industrializing* countries has helped reveal disparities across the globe: many industrialized countries that have adopted asbestos bans still have a substantial burden of ARD, while many industrializing countries continue to use asbestos at substantial levels. However, it is unclear whether these industrializing countries do not actually have a substantial ARD burden or if they are unable to identify their burdens. In pursuit of economic gains, these industrializing countries show little incentive to adopt bans despite the availability of viable substitutes for asbestos. Such situations can be viewed as reflecting the national conditions of resource-poor countries. Nonetheless, if the world is to prevent repeating the tragic experience of industrialized countries, more insight must be gained regarding the global and national status of asbestos and ARD in light of the economic status of countries.

Malignant mesothelioma (herein called mesothelioma) is a rare but a fatal form of carcinoma (herein called cancer) that is specifically caused by asbestos exposure. Many studies have estimated the mesothelioma disease burden from mortality data because about half of those diagnosed with mesothelioma die within a year [3]. The disease burden of mesothelioma has been the subject of scientific analyses at global [4,5], regional [6,7], and country levels [8,9]. Recently, many studies using the estimated global and country-level mesothelioma burdens have referenced the Global Burden of Disease (GBD) studies [10,11,12] (C-OO, TD, JT, and KT are GBD contributors), which is widely accepted as a reliable source of data for global health. The GBD studies assess the burdens of many diseases, including mesothelioma. Separately, Odgerel et al. estimated the global mesothelioma death burden [13].

Both Odgerel et al. [13] and the GBD [12] referenced existing national mortality data to derive global estimates of the mesothelioma burden at 38,000 and 30,000 annual deaths, respectively. Subsequent updates moderately increased the GBD estimates on mesothelioma. The estimates of the two studies were strikingly different for individual countries; for example, the estimated mesothelioma burdens for China and India were 3.8- and 2.5-fold higher, respectively, in Odgerel et al. versus GBD. These countries are classified as upper middle-income (UMI) and lower middle-income (LMI) countries, respectively, according to the World Bank [14]. In general, disease burden estimates for lower income countries, relative to those for high-income countries, have a larger effect on the global burden because of their population size. It is thus pertinent to cross-examine estimates of the global mesothelioma burden stratified by the national income status of countries. Moreover, it is important to address national income gaps regarding data availability and quality with respect to asbestos and ARD. The findings of such analyses should provide guidance for national and international policies aimed at improving the relevant situations of lower-income countries.

The objectives of this study are therefore to compare estimates of the global-level mesothelioma burden focusing on how existing national mortality data are utilized, and further, to assess the interrelationship of country-level mesothelioma burdens and asbestos use with national income.

## 2. Materials and Methods

Mesothelioma was defined as International Classification of Diseases (ICD) code C45. For our comparative overview, we extracted global mesothelioma death burdens from various sources, including individual articles and the series of articles published as the Global Burden of Disease (GBD) studies (Table 1).

As an indicator of national income status, we applied the 2019–2020 World Bank “National Income Categories (NIC)” of “high-income (HI),” “upper middle-income (UMI),” “lower middle-income (LMI)”, and “low income (LI)”, which were available for 218 countries. As the size of population (N = 776,000) and number of countries in the LI category (N = 31) was small, we absorbed the LI category into the LMI category. Odgerel et al. had previously derived country-level mesothelioma death burdens for 230 countries based on data extracted from the WHO Mortality Database as of November 2015. However, because the present study focused on national income status as of 2019–2020, our analyses was restricted to the abovementioned 218 countries, which was a subgroup of the previously studied 230 countries.

In the present study, we updated country-level mesothelioma death burdens to the most recent status based on data extracted from the WHO Mortality Database as of December 2019 (Table 2 and Figure 1) [15]. Specifically, data on mesothelioma deaths from 1994 to 2018 were extracted from the WHO Mortality Database as of December 2019. Countries were defined as “reporting” (or “nonreporting”) if data on mesothelioma were identified (unidentified) for the country. Of the 109 countries reporting mesothelioma deaths to WHO as of December 2019, five countries (French Guyana, Guadeloupe, Martinique, Mayotte, and Reunion) were excluded from further analysis due to lack of information on NIC. For data identified as mesothelioma, we applied previously described criteria to distinguish “reliable” and “unreliable” data [13] by determining if: (1) a crude mortality rate for the entire period of 1994–2018 was 0.5 death per million per year or less, which is half the 1.0 case per million per year that is widely accepted as the background level of mesothelioma [16,17] (i.e., strongly suggesting that the country suffered from underreporting and/or underdiagnosis; “underreporting” hereafter); (2) there were two or fewer reporting years; and/or (3) there were 10 or fewer total reported deaths. If a country satisfied one or more of these conditions, it was considered “unreliable”, otherwise, it was considered to have “reliable” data.

Odgerel et al. derived their best estimate of the country-level mesothelioma death burden by extrapolating the gender- and age-specific death rates of mesothelioma to the population structure of nonreporting countries categorized based on their level of historical asbestos use [13]. In the current study, the same algorithm was applied but used categories of national income (HI, UMI, and LMI) to estimate the missing mesothelioma death burdens. These estimates were further compared to the GBD 2017 estimates of the mesothelioma death burden categorized by national income status [18].

The numbers published by Odgerel et al. for mesothelioma death burdens based on data extracted from the WHO Mortality Database as of November 2015 were herein reused for two purposes: (1) to compare the mesothelioma death burden of Odgerel et al. with that of the GBD study stratified by national income status (Table 3); and (2) to rank country-level mesothelioma burdens grouped by national income status (Figure 2).

Data on asbestos use were extracted from the US Geological Survey (USGS) [19,20,21]. We adopted a definition of use (production + import - export) and obtained data by country from across various time intervals between 1920 and 2016. Asbestos use in countries that made a political transition was treated as previously described [22].

Microsoft Excel (Microsoft Corporation, Redmond, WA, USA) and SAS Version 9.4 (SAS Institute, Inc., Cary, NC, USA) were used to organize, compile, and analyze all data.

## 3. Results

Table 1 shows the data sources for the estimated numbers of global mesothelioma deaths. “Malignant mesothelioma” was first introduced into the tenth revision of the ICD-10 in 1994 [23]. Since then, a range of WHO member states have reported the number of deaths caused by mesothelioma to the WHO. The number of “reporting” countries has grown over the years, but today it still represents about half the WHO member states (104 of all 194 member states as of December 2019). Because underdiagnosis and underreporting of mesothelioma are widespread, the reported numbers are not always “reliable”; this is especially true for countries with few cases. The numbers of global mesothelioma deaths have been estimated using a variety of algorithms that account for the reported numbers in different ways.

From 1990 [24] to 2017, the GBD studies estimated global mesothelioma deaths within the range of 18,200 to 32,400; however, the estimate of 43,000 by Driscoll et al. [25] has been most widely referenced, including by the WHO [26]. Note that mesothelioma is one of the many causes of deaths that are estimated in the GBD studies. Separately, the group led by Takahashi has been documenting reported and/or estimated numbers of global mesothelioma deaths in studies that focused specifically on country-level data on mesothelioma and/or asbestosis. These articles further drew implications on future projections based on the national mortality trends and patterns of asbestos use by countries and suggested that a rise in the global mesothelioma deaths is inevitable, especially in developing countries [4,22,27]. Odgerel et al. [13] reported a global estimate of 38,400 annual deaths by summing reported numbers that the authors judged as “reliable” with estimated numbers for countries judged as having unreliable data or countries that did not report data.

Globocan 2018, which is an online database that provides estimates of incidence and mortality in 185 countries for 36 types of cancer as part of IARC’s Global Cancer Observatory [28], reported 25,576 mesothelioma deaths in 130 UN Member States by using data from the WHO Mortality Database for some countries and applying different models to estimate numbers for other countries.

Table 2 shows that the country status of reporting mesothelioma deaths varied widely by national income category. Among 80 high-income (HI) countries, 54 (68%) reported mesothelioma to the WHO and 26 (32%) did not, and among 60 upper middle-income (UMI) countries, the respective numbers (proportions) were 39 countries (65%) and 21 countries (35%), respectively. In contrast, among 78 lower middle-income and low-income (LMI) countries, only 11 (14%) reported mesothelioma deaths while 67 (86%) did not. Of all 218 countries, 104 countries (48%) were designated as reporting and 114 countries (52%) were designated as nonreporting. The majority of the reporting 104 countries were HI countries (52%, or 54 countries), while the majority of the 114 nonreporting countries were LMI countries (59%, or 67 countries).

Table 3 compares the GBD [12,18] and Odgerel et al. [13] studies in terms of their estimated numbers of global mesothelioma deaths according to three country groups categorized by national income. Global mesothelioma deaths were estimated at about 30,000 and 38,000 by GBD and Odgerel et al., respectively. The GBD and Odgerel et al. studies respectively estimated that 58% and 37% of global mesothelioma deaths arise from HI countries, while 25% and 41% arise from UMI countries. It should be noted that the estimation methods differed between the two studies: the GBD estimated all data (estimation was based on reported data taking into account the perceived accuracy of data), while Odgerel et al. combined reported numbers (that authors judged as reliable) with estimated numbers. In the latter study, almost all (98%) of the numbers were reported numbers in HI countries, whereas 92% and 99.9% of all numbers were estimated numbers for UMI and LMI countries, respectively. Importantly, the GBD estimated number of mesothelioma deaths for HI countries (17,460) is nearly 3000 higher than that of Odgerel et al., with 98% of the latter comprising reported numbers.

Figure 1 shows the trend over time in number of countries reporting mesothelioma deaths to the WHO reanalyzed from Odgerel et al. The reporting countries comprise mostly HI countries at about 30–55 countries per year, followed by UMI countries at about 20–40 countries per year and LMI countries at about 10 countries per year during 1994–2018. The increase in the number of reporting countries over time is most apparent for HI countries followed by UMI countries, with the least change seen for LMI countries. The most recent drop in the number of countries in all income categories reflects the delay of countries in reporting data. The difference across income groups is most apparent in the proportion of countries that reported reliable data: whereas most HI countries reported reliable data, this was true for only about half of UMI countries and only one LMI country.

Figure 2 shows the annual number of reported and/or estimated mesothelioma deaths by countries grouped by national income category. Mesothelioma deaths were reported and “reliable” in the majority of HI countries, led by the US, the UK, Italy, Germany, and Japan, at about 2600, 2400, 1400, 1400, and 1400 reported annual mesothelioma deaths, respectively. Only 18 UMI countries reported “reliable” data, led by Turkey at about 360 annual mesothelioma deaths. In the remaining 42 UMI countries, mesothelioma deaths were estimated (unreliable or no data). For example, China, Russia, and Brazil were estimated to have 10,000+, about 1300, and about 1200 annual mesothelioma deaths, respectively. In the LMI category, only the Republic of Moldova reported “reliable” data at four mesothelioma deaths per year. The estimated number was highest for India at nearly 6000, followed by Indonesia at about 550 mesothelioma deaths per year. For each national income category, the numbers of reported/estimated annual mesothelioma deaths were highly skewed, with 85%–96% incurred by the top ten countries.

Figure 3 shows the asbestos consumption during 1920–2016 in the same three country groups. Global asbestos use surpassed 2.0 million metric tonnes in 1960, peaked around 1980 at about 5.0 million metric tonnes, and started to decrease thereafter. Following the turn of the century, total consumption remained stable at about 2.0 million metric tonnes until around 2013, after which, consumption declined further to 1.4 million in 2016. By country group, HI countries consumed the bulk of the global volume until around 1970; they then reduced their share and were overtaken by UMI countries, which began consuming the largest share of the global volume around 1985. In more recent years, the LMI countries have increased their share and volume of asbestos use, closing the gap with UMI countries. Since 2010, global asbestos consumption has been shared by only the UMI and LMI countries.

Figure 4 shows the cumulative amount of asbestos use since 1980 (the peak year of global use) by the top ten users from the HI, UMI, and LMI country groups. Among them, asbestos use has been banned in eight, one, and zero of the HI, UMI, and LMI countries, respectively. Among the HI countries, only Japan exceeded 1 million metric tonnes of cumulative asbestos use, while the other nine countries used between 100,000 and 1 million metric tonnes. Among the UMI countries, five countries exceeded 1 million metric tonnes of cumulative asbestos use (Russia—11.5, China—10.8, Brazil—3.6, Kazakhstan—3.2, and Thailand—2.4 million), while the other five countries used between 300,000 and 1 million metric tonnes. Among the LMI countries, four countries exceeded 1 million metric tonnes of cumulative asbestos use (India—5.7, Indonesia—1.6, Ukraine—1.4, Vietnam—1.0 million), while the other six countries used between 85,000 and 1 million metric tonnes.

## 4. Discussion

The current study compared several information sources that offered estimates on the global burden of mesothelioma deaths. Studies by individual author groups generally reported estimates of about 40,000 annual deaths and thus were higher than the GBD estimates of about 30,000 annual deaths. Recent GBD studies have reported moderately higher numbers than before. The Odgerel et al. study judged the reliability of reported data and added the “reliable” reported data to estimated numbers for countries judged to have “unreliable” or no data. In contrast, the GBD studies estimated all data (estimation was based on reported data taking into account the perceived accuracy of data). Our findings indicate that the national income status was related to the reporting status of mesothelioma deaths: the majority of HI countries reported mesothelioma deaths to the WHO, whereas the majority of LMI countries did not. The national income status was also found to be related to the estimated burden of mesothelioma deaths, the historical asbestos consumption, and the asbestos ban status. The implication is that although HI countries currently shoulder the bulk of global mesothelioma deaths, this will eventually shift to the lower income (UMI and LMI in this study) countries.

The present study found that approximately half of all countries reported mesothelioma deaths to the WHO. Of these “reporting” countries, more than half, more than a third, and only one-tenth corresponded to HI, UMI, and LMI countries, respectively. Conversely, about two thirds of HI (68%) and UMI (65%) countries reported mesothelioma deaths, whereas the majority (86%) of LMI countries did not. The low reporting rates should be viewed against the general consensus that there is a “background” level of mesothelioma of about 1 per million population per year, which suggests that a country with a population of 1 million is likely to incur at least one mesothelioma death per year. Mesothelioma remains a difficult disease to diagnose, and pleural mesothelioma is frequently misdiagnosed for lung cancer. The pathological confirmation of a mesothelioma diagnosis requires the proper use of a panel of immunohistochemical markers, and this option is often unavailable in resource-poor countries. The gradient in the reporting status of mesothelioma by national income thus reflects the gradient in the level of medical resources and, in particular, their absence in LMI countries.

The quality of data on mesothelioma deaths is influenced by the quality of healthcare combined with a well-functioning registration system for causes of death. The registration system for causes of death becomes incrementally substandard in lower-income countries. In recent decades, data reporting has improved in HI and UMI countries, but little has changed in LMI countries. Even if data are reported, it is widely accepted that little of these reported data contribute to data analysis or decision-making because of their poor quality, especially in LMI countries [29,30]. Determining the quality of health data is a complex undertaking because data quality encompasses multiple dimensions, including accuracy, completeness, timeliness, and consistency [31]. For the mesothelioma mortality data reanalyzed from the original Odgerel et al. study, we applied three criteria (for details, see Methods) to judge their “reliability.” After we excluded “unreliable” data, only half of reporting countries were considered to have “reliable” data. When we applied objective criteria, the majority of HI countries were judged as reporting “reliable” data, whereas 90% of LMI countries were determined to be reporting “unreliable” data; these determinations led to their inclusion in or exclusion from the analysis, respectively.

Our comparison of the estimates reported by the GBD and Odgerel et al. on mesothelioma deaths revealed considerable differences in the total global estimates and the numbers estimated across national income categories (Table 3). Whereas the GBD attributed the highest proportion (nearly 60%) of global mesothelioma deaths to HI countries, Odgerel et al. attributed the highest proportion (40%+) to UMI countries, followed by HI countries (<40%). As mentioned earlier, the GBD study estimated numbers for all countries, whereas Odgerel et al. combined reported and estimated numbers. In effect, the GBD estimate of 17,460 for HI countries implied that their reported number was “underreported” by 3000, whereas Odgerel et al. accepted the number reported by HI countries as “reliable.” A further difference is that, relative to the GBD study, Odgerel et al. estimated a more than two-fold higher number for UMI countries and 3000+ more for LMI countries, which resulted in their overall estimate being 8000 higher (at 38,000). Therefore, we assume that the magnitude of underdiagnosis and underreporting of mesothelioma likely exceeds that of overdiagnosis and overreporting, although the resulting underestimation gap should be narrowing in HI countries. The critical part of estimation thus lies with the UMI and LMI countries.

Studies to estimate disease burdens typically incorporate some form of extrapolation and rely heavily on the availability and quality of reported data. Mesothelioma presents many challenges in general and for lower-income countries in particular, such as rareness of disease, shortage of histopathologists, and insufficient capacity to conduct immunohistochemistry diagnostic analysis [32,33,34]. Underreporting of mesothelioma as an ARD is rampant; this may be unintentional due to lack of technology and medico-social infrastructure or intentional due to sociopolitical disincentives. The present study confirmed that lower-income countries seriously lacked data, let alone “reliable” data, on the mesothelioma death burden. This lack of information is not limited to the reported numbers of mesothelioma; it is also reflected in the general information on ARD. Lin et al. recently reported that more than 80% of all scientific articles related to ARD are produced by only ten HI countries, and almost none is produced by lower-income countries [35]. The margin of error for estimated numbers should thus be incremental, ranging from the smallest in HI countries to the largest in LMI countries.

The global volume of asbestos consumption peaked in 1980, declined rapidly between 1980 and 1995, remained stable until around 2012, and thereafter declined further. The rapid decline around 1980–1995 may be attributed to shifts first in HI countries and then in UMI countries. In contrast, LMI countries have increased their volume and share in recent years. Since the last decade, almost no consumption has been recorded by HI countries, meaning that the global asbestos consumption is now shared solely by UMI and LMI countries. The national trends in asbestos use are closely related to the national status of asbestos bans, although there are numerous contributing factors. For example, many HI countries started to reduce asbestos use *before* officially adopting bans, primarily due to concerns over public health. On the other hand, the US over a considerable historical time recorded the highest consumption in the world but has now reduced its use to a low (albeit nonzero) level without officially banning asbestos.

A few exceptional countries notwithstanding, the global situation is thus a mixed bag of dilemmas. The HI countries initially used large amounts of asbestos but then experienced a surge of mesothelioma deaths and adopted asbestos bans; however, the mesothelioma burden still lingers today. The lower-income countries started to use asbestos later and either have not yet seen the mesothelioma burden or have not yet detected the true burden due to technological limitations. This invisibility may contribute to the continued use of asbestos in such countries. However, the amount of asbestos used by countries is a strong predictor of the mesothelioma burden that they will eventually shoulder [36]. In addition, accumulating evidence indicates that asbestos bans reduce national mesothelioma burdens [37,38]. As industrial hygiene measures to minimize occupational exposure to asbestos are known to be lacking in lower-income countries, the future implications for asbestos-using lower-income countries are grim.

In each national income category, the number of mesothelioma deaths was highly skewed, with around 90% incurred by the top ten countries. This is largely due to the skewed size of the national populations, which in turn is related to the skewed amount of cumulative asbestos use. Three countries with large populations and known high asbestos use are likely to impact the global mesothelioma burden: China, Russia, and India. The estimates by Odgerel et al. and GBD 2017 for China are 10,460 and 2780 mesothelioma deaths, or 8.0 and 2.1 per million (M) population, respectively. Previously reported estimates range from 1.5 per M [39] and 2–3 per M [40] to 9.1 per M [41]. The estimate of Zhou et al. [41] is equivalent to 11,860 mesothelioma cases/deaths, and thus comes close to that of Odgerel et al. Note that China used asbestos at 207,000 metric tonnes per year during 1970–1980 (the most relevant period for recent mesothelioma). Estimates for Russia are 1292 and 611 mesothelioma deaths, or 8.7 and 4.2 per M, respectively. Kaprina [42] pointed out that the mortality statistics published by the Russian government fail to separate the number of mesothelioma deaths from the combined category of “neoplasms of mesothelial and soft tissues (C45–C49)”. Most Russian scientific articles on mesothelioma are clinically oriented, and there is little epidemiological data. Previously reported estimates range from 2 per M in men and 1 per M in women [43] to 15–20 per M in men and 3 per M in women [44]. Note that Russia used asbestos at 840,000 metric tonnes per year during 1970–1980 (the most relevant period for recent mesothelioma). Estimates for India are 5940 and 2422 mesothelioma deaths, or 4.5 or 1.8 per M, respectively. For India, the ARD burden including mesothelioma is “inevitable,” but the scale of the problem is unclear due to the existence of a “poor, almost nonexistent, system to record death and disease” [45].

The present study has some limitations. First, it combined a reanalysis of previously reported data (national status of reporting mesothelioma as of November 2015) with a new analysis of new data on mesothelioma deaths (as of December 2019); the two sets of data had an interval of four years. Second, the national income category reflected only the most recent economic status of each country. As the national economic status can change over time, employing a “cross-sectional” indicator to group countries will bias findings. However, the categories HI, UMI, and LMI (the latter combining lower middle-income and low-income) were broad; the change in the “relative” economic status across countries should be limited and thus bias should be minimized. Third, we used the amount of raw asbestos consumption as an indicator of asbestos use. An ideal macro indicator should include asbestos-containing products (ACP) in addition to raw asbestos. However, many countries manufacture construction-purpose ACP (e.g., asbestos-cement roofs) from raw asbestos. Additionally, raw asbestos can serve as a surrogate indicator of asbestos use, assuming that imports of ACP are negligible. Future studies should address this issue more adequately. A strength of the present study was that it offers a systematic comparison of recently reported estimates of the global mesothelioma burden, including those by a highly referenced source (the GBD study); this allowed us to undertake a detailed examination of the availability and reliability of relevant data and information, which we found to be incrementally poorer among countries of lower national income.

## 5. Conclusions

The global mesothelioma burden, based on reported numbers, is currently shouldered predominantly by the HI countries. Estimations have been conducted because reliable data are lacking in UMI countries and totally absent in LMI countries, where large margins of error on estimates are to be expected. Based on recorded historical, recent, and current use of asbestos, however, mesothelioma burdens will likely manifest soon in UMI and eventually in LMI countries. HI countries should work with lower-income countries to enable effective sharing of the technologies and infrastructure to develop and use cheap alternatives to asbestos and to diagnose and control mesothelioma. Most importantly, all countries should accept the WHO statement that “the most efficient way to eliminate asbestos-related diseases is to stop using all types of asbestos” as a *de facto* mandate to ban asbestos.

## Figures and Tables

**Figure 1 ijerph-17-06900-f001:**
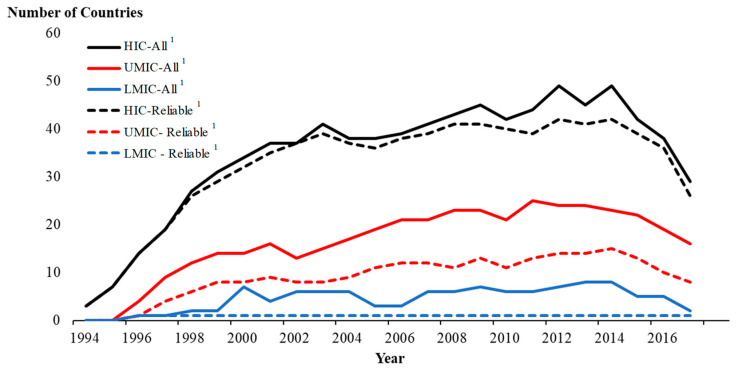
Trend over time in number of countries reporting mesothelioma deaths to the WHO by national income category ^1^ and data reliability ^2^.HIC: High income countries; UMIC: Upper middle-income countries; LMIC: Lower middle- and low-income countries combined (See Table 2 and text for exact definition of national income category). ^1^ All countries reporting mesothelioma deaths to the WHO. ^2^ Of ^1^, countries reporting data judged to be “reliable” according to Odgerel et al. [13].

**Figure 2 ijerph-17-06900-f002:**
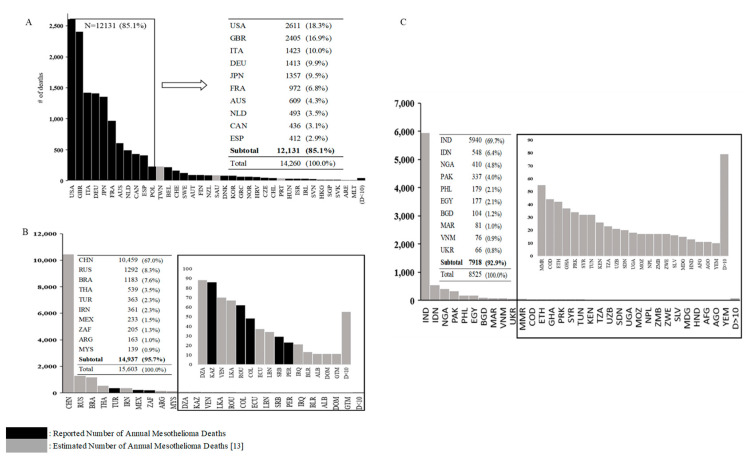
Number of Reported/Estimated Annual Mesothelioma Deaths by Countries According to National Income Category. A: High income countries (N = 80), B: Upper middle-income countries (N = 60); C: Lower middle income and low-income countries combined (N = 78). See Table 2 and text for exact definition of national income category. Estimated numbers are from Odgerel et al 2017 [13], which the present study (for the first time) grouped by national income category. Countries are listed by three-letter codes, e.g., USA for the United States. Appendix A lists all codes used. “D < 10” indicates the group of countries with fewer than 10 deaths (either reported or estimated) for which the aggregate number of deaths is shown. Appendix A lists these countries.

**Figure 3 ijerph-17-06900-f003:**
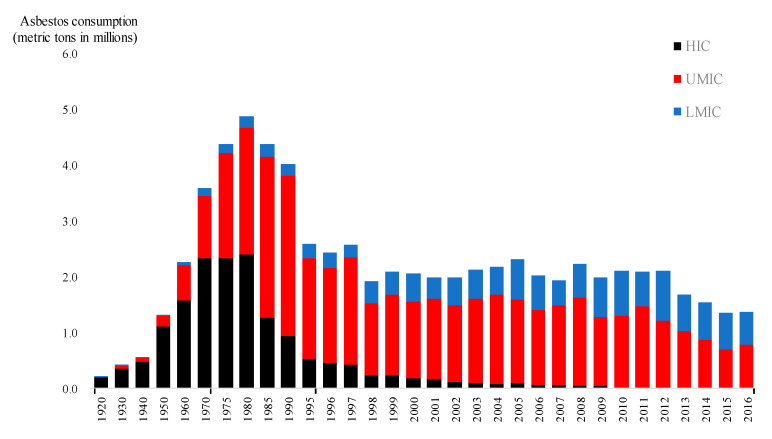
Global Trend Over Time in Asbestos Use of the 3 Country Groups based on National Income Category. HIC: High income countries (N = 54), UMIC: Upper middle-income countries (N = 51); LMIC: Lower middle income and low-income countries combined (N = 57). See Table 2 and text for exact definition of national income category. USGS reported world asbestos consumption with 10- and 5- year interval during 1920–1970 and 1975–1995, respectively, and every single year since 1995.

**Figure 4 ijerph-17-06900-f004:**
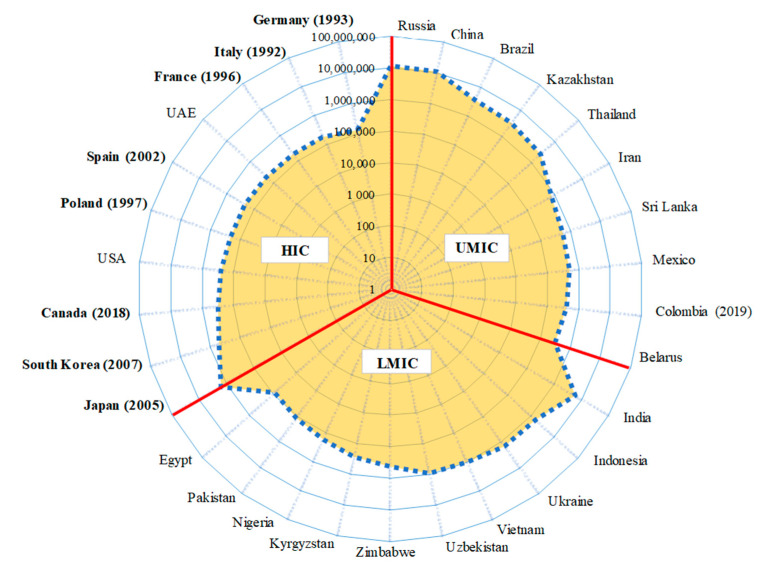
Cumulative Use of Asbestos (metric tonnes) Since 1980 by Country and National Income Category. HIC: High income countries, UMIC: Upper middle-income countries; LMIC: Lower middle income and low-income countries combined. See Table 2 and text for exact definition of national income category. Only top 10 countries in each national income category that recorded the highest cumulative asbestos use since 1980 are shown. Countries that have banned asbestos are shown in bold with the year of banning in brackets.

**Table 1 ijerph-17-06900-t001:** Data sources for estimated numbers of global mesothelioma deaths.

Data Sources	Mesothelioma Deaths	Implications for Future Projections
**Journal Articles**		
Driscoll (2005) [25]	43,000 annual deaths in the world	Not applicable
Delgermaa (2011) [4]	92,253 reported deaths in 83 WHO member states (during 1994–2008)	“*Since asbestos use has recently increased in developing countries, a corresponding shift in disease occurrence is anticipated in developing countries*”
Park (2011) [22]	38,900 estimated deaths in 33 countries (during 1994–2008)	“*Particularly since 1970 onward, these countries should anticipate the need to deal with a very high burden of mesothelioma in the immediate decades ahead*”
Diandini (2013) [27]	11,884 reported annual deaths in 82 countries (1994–2010)	Not applicable
Odgerel (2017) [13]	38,400 annual deaths (sum of estimated and reported) in the world (during 2008 and 2014)	“*The estimation of the global mesothelioma burden will naturally improve as more countries report quality data and fewer countries require estimates to be made*”

**Global Burden of Disease (GBD) estimation**		
GBD (1990) [24]	18,213 estimated annual deaths in the world	Not applicable
GBD (1995) [24]	19,318 estimated annual deaths in the world	Not applicable
GBD (2000) [24]	21,367 estimated annual deaths in the world	Not applicable
GBD (2005) [24]	23,193 estimated annual deaths in the world	Not applicable
GBD (2010) [24]	27,849 estimated annual deaths in the world	Not applicable
GBD (2015) [24]	32,372 estimated annual deaths in the world	Not applicable
GBD (2017) [12]	29,908 estimated annual deaths in the world	Not applicable

**Global Databases**		
WHO MDB (2019) [15]	13,993 deaths in 104 WHO member states	Not applicable
WHO Globocan (2018) [26]	25,576 deaths in 130 UN Member States	Not applicable

Reference numbers match with reference numbers in main text

**Table 2 ijerph-17-06900-t002:** Status of countries’ reporting of mesothelioma deaths to the WHO by national income category^1^.

Countries Grouped by National Income Category ^1^	Reporting ^2,3^		Nonreporting ^2^		Total	
High income (HI)	54	(52%)	26	(23%)	80	(37%)
	[68%]		[32%]		[100%]	
Upper middle income (UMI)	39	(38%)	21	(18%)	60	(28%)
	[65%]		[35%]		[100%]	
Low middle income (LMI) ^4^	11	(10%)	67	(59%)	78	(35%)
	[14%]		[86%]		[100%]	
Total	104	(100%)	114	(100%)	218	(100%)
	[48%]		[52%]		[100%]	

^1^ 218 countries were categorized according to the World Bank: “countries classification by income level: 2019–2020”, ^2^ Above countries were further grouped by either reporting or nonreporting of mesothelioma deaths to the WHO during 1994–2018. ^3^ 109 countries reported mesothelioma death to WHO as of December 2019; five of them (French Guyana, Guadeloupe, Martinique, Mayotte, and Reunion) were not categorized by income level in 1), thus excluded from analysis. ^4^ LMI and low income (LI) combined.

**Table 3 ijerph-17-06900-t003:** Distribution of global mesothelioma deaths across countries by national income category ^1.^

Source	HI		UMI		LMI		Total	
GBD (2017) [12]	17,460	(58%)	7358	(25%)	5090	(17%)	29,908	(100%)

Odgerel (2017) [18]	14,251	(37%)	15,603	(41%)	8534	(22%)	38,388	(100%)
	[100%]		[100%]		[100%]		[100%]	
Reported to the WHO	13,920	(92%)	1252	(8%)	4	(0%)	15,176	(100%)
	[98%]		[8%]		[0.1%]		[40%]	
Estimated	331	(1%)	14,351	(62%)	8530	(37%)	23,212	(100%)
	[2%]		[92%]		[99.9%]		[60%]	

^1^ 218 countries were categorized according to the World Bank: “countries classification by income level: 2019–2020”; HI: High Income; UMIC: Upper Middle Income; and LMIC: Lower Middle Income and Low Income combined. Reference numbers match with reference numbers in main text.

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
