# Peer review of "Burden of Mesothelioma Deaths by National Income Category: Current Status and Future Implications"

_ijerph, 2020, doi:10.3390/ijerph17186900_

Round 1
Reviewer 1 Report
This is a well written informative paper, the English grammar is impeccable. For some reasons the evaluation sheet does not me allow to choose that the english language is fine and I was forced to check minor spell requirement. I have only couple of suggestions
The Authors note: "The magnitude of underdiagnosis and underreporting of mesothelioma likely exceeds that of overdiagnosis and overreporting,..." Regardless of whether there is more under than over reporting, the bottom line is that given the problems with reliable diagnosis we can only guess about the true incidence of mesothelioma globally. I think this should be clearly stated in the abstract. In other words the statement in the abstract "the global mesothelioma burden is currently shouldered predominantly by high-income countries" is an hypothesis more than a fact, since obviously we have no idea about the true incidence of mesothelioma in low income countries, such as India for example that consumes increasing amounts of asbestos each year. See discussion in ref 5.
Secondly I disagree with the characterization that "mesothelioma is difficult to diagnose". Mesothelioma is easy to diagnose for a trained pathologist who has a supporting immunohistochemical and possibly EM laboratory. Of course some cases present challenges, especially that rare spindle cell mesotheliomas (sarcomatoid) but in general an epithelial and biphasic mesothelioma should be easily diagnosed by any experienced pathologist. However, mesothelioma is difficult to diagnose when there is no supporting immunohistochemistry and electron microscopy, which is the norm in low income countries. I suggest to clarify this issue.
Author Response
Thank you for the valuable comments in reviewing our manuscript. The following is our point-by-point responses which were reflected in the revised version of the manuscript. The manuscript has been revised using track changes and both clean and marked copies have been submitted.
Author’s note:
Reviewer #1:
Comment 1: "The magnitude of underdiagnosis and underreporting of mesothelioma likely exceeds that of overdiagnosis and overreporting,..." Regardless of whether there is more under than over reporting, the bottom line is that given the problems with reliable diagnosis we can only guess about the true incidence of mesothelioma globally. I think this should be clearly stated in the abstract. In other words the statement in the abstract "the global mesothelioma burden is currently shouldered predominantly by high-income countries" is an hypothesis more than a fact, since obviously we have no idea about the true incidence of mesothelioma in low income countries, such as India for example that consumes increasing amounts of asbestos each year. See discussion in ref 5.
Our response:
We agree that true global burden of mesothelioma is speculative. However, it is also true that, more estimations are conducted (e.g., GBD series of studies and the study by Odgerel et al) in order to better estimate the actual situation and to narrow the margin of error. Indeed, accuracies of estimates are being calibrated against reported values which opportunities increase over the years. However, the quality of reported values differs across countries which we presumed are related to national income status. Our paper addressed this issue of estimated versus reported numbers by national income status. Please refer to the following text (Line 295-300): “In effect, the GBD estimate of 17,460 for HI countries implied that their reported number was “underreported” by 3,000, whereas Odgerel et al. accepted the number reported by HI countries as “reliable.” A further difference is that, relative to the GBD study, Odgerel et al. estimated a more than two-fold higher number for UMI countries and 3,000+ more for LMI countries, which resulted in their overall estimate being 8,000 higher (at 38,000).”
In other words the statement in the abstract "the global mesothelioma burden is currently shouldered predominantly by high-income countries" is an hypothesis more than a fact, since obviously we have no idea about the true incidence of mesothelioma in low income countries, such as India for example that consumes increasing amounts of asbestos each year.
Our response: We accepted the comment and clarified (narrowed) the meaning of our statement in the abstract (Line 34), and in the conclusion (Line 377), by inserting the phrase “based on reported numbers.”
Comment 2: Secondly I disagree with the characterization that "mesothelioma is difficult to diagnose". Mesothelioma is easy to diagnose for a trained pathologist who has a supporting immunohistochemical and possibly EM laboratory. Of course some cases present challenges, especially that rare spindle cell mesotheliomas (sarcomatoid) but in general an epithelial and biphasic mesothelioma should be easily diagnosed by any experienced pathologist. However, mesothelioma is difficult to diagnose when there is no supporting immunohistochemistry and electron microscopy, which is the norm in low income countries. I suggest to clarify this issue.
Our response: We have reflected the reviewer’s comment and revised the text as follows: “Mesothelioma presents many challenges in general, and for lower-income countries in particular, such as rareness of disease, shortage of histopathologists, and insufficient capacity to conduct immunohistochemistry diagnostic analysis.” (Line 305-307)

Reviewer 2 Report
- The authors collected data from different sources from different time periods:
Data on asbestos use were extracted from the US Geological Survey and obtained data by country from across 120 various time intervals between 1920 and 2016
National income status, we applied the 2019-2020 World Bank “National
84 Income Categories (NIC)”
Data on mesothelioma deaths from 1994 to 2018 were extracted from the 95 WHO Mortality Database as of December 2019.
National income status for countries change significantly from 1920 to 2016, so how can the income data from 2019-20 be compared to the asbestos data from 1920 to 2016 and then correlated with the mesothelioma deaths from 1994 t0 2018? Please clarify this comparison, as that is the biggest limitation of the study.
2. There was no randomization or a control group, therefore the authors are not in a position to generalize the study or talk about causation. They have only demonstrated a relationship. This should be explicitly mentioned in the conclusion section.
3.Please have the manuscript reviewed for English grammar and editing. The use of words “we” and “our” should be replaced by “this study” or “this research”. Numerous language flaws and grammar concerns with the manuscript. Please rewrite/have to thoroughly reviewed by an English language expert.
Author Response
Thank you for the valuable comments in reviewing our manuscript. The following is our point-by-point responses which were reflected in the revised version of the manuscript. The manuscript has been revised using track changes and both clean and marked copies have been submitted.
Author’s note:
Reviewer #2:
Comment 1: The authors collected data from different sources from different time periods:
Data on asbestos use were extracted from the US Geological Survey and obtained data by country from across 120 various time intervals between 1920 and 2016
- National income status, we applied the 2019-2020 World Bank “National
84 Income Categories (NIC)”
- Data on mesothelioma deaths from 1994 to 2018 were extracted from the 95 WHO Mortality Database as of December 2019.
National income status for countries change significantly from 1920 to 2016, so how can the income data from 2019-20 be compared to the asbestos data from 1920 to 2016 and then correlated with the mesothelioma deaths from 1994 t0 2018? Please clarify this comparison, as that is the biggest limitation of the study.
Our response: We agree broadly to this criticism including the fact that the national income per se (values in monetary terms) changes significantly. In contrast, however, the national income category (in terms of HI, MI, LI), and hence the relative “standing” between countries changes to a limited extent. Moreover, data on national income categories are not available for the earlier years. It is not uncommon to apply the most recent national income category to group countries in analyzing historical trends (See Le et al. National Use of Asbestos in Relation to Economic Development. Environ Health Perspect 2010; 118(1): 116-119). However, we recognize this problem as a limitation, and stated as such (Line 362-366) that “the national income category reflected only the most recent economic status of each country. As the national economic status can change over time, employing a “cross-sectional” indicator to group countries will bias findings. However, the categories HI, UMI and LMI (the latter combining lower middle-income and low-income) were broad; the change in the “relative” economic status across countries should be limited and thus bias should be minimized.”
Comment 2: There was no randomization or a control group, therefore the authors are not in a position to generalize the study or talk about causation. They have only demonstrated a relationship. This should be explicitly mentioned in the conclusion section.
Our response: The current study did not attempt to establish causation. The study is fundamentally a descriptive analysis for which randomization or control group is not warranted. We drew implications based on findings from the descriptive analysis which were discussed logically in combination with existing knowledge from a range of references. In an effort to accommodate this comment, we have toned down our statement in the abstract and the Conclusion (Line 381) by inserting “likely” into the phrase “burdens will manifest.”
Comment 3: Please have the manuscript reviewed for English grammar and editing. The use of words “we” and “our” should be replaced by “this study” or “this research”. Numerous language flaws and grammar concerns with the manuscript. Please rewrite/have to thoroughly reviewed by an English language expert.
Our response: The manuscript was edited and proofread by English Manager Science Editing LLC (https://www.sciencemanager.com/About_Us.htm). Please also note that REVIEWER #1 stated that “the English grammar is impeccable. For some reasons the evaluation sheet does not me allow to choose that the English language is fine and I was forced to check minor spell requirement.” However, we accommodated the reviewer’s comment by making some replacements for the words “we” and “our.”

Reviewer 3 Report
This is a very well written paper with significant relevance for the researchers interested in evaluate and understand the incidence and rate of death of mesothelioma globally. The proposal to evaluate disease burden in terms of the countries income status was interesting and well sustained.
In order to improve the manuscript, some specific suggestions are recommended:
- Line 80:Consider to include the meaning of ICD at the begin of this paragraph: "Mesothelioma was defined as an International Clasificcation Diseases (ICD) code 45."
- Please, confirm that this sentence is correct, it looks to me that is contradictory: (Line 264-265) ..."Conversely, about two thirds of HI and UMI countries and less than two thirds of UMI countries reported mesothelioma deaths, ..."
- If disease burden estimates for lower income countries have a larger effect on the global burden due to their population size, why not consider to extrapolate the number of inhabitants per country or its growth population rate to estimate of the country-level mesothelioma death burden?
-
Author Response
Thank you for the valuable comments in reviewing our manuscript. The following is our point-by-point responses which were reflected in the revised version of the manuscript. The manuscript has been revised using track changes and both clean and marked copies have been submitted.
Author’s note:
Reviewer #3:
Comment 1: Line 80: Consider to include the meaning of ICD at the begin of this paragraph: "Mesothelioma was defined as an International Classification Diseases (ICD) code 45."
Our response: We have done as suggested. (Line 80)
Comment 2. Please, confirm that this sentence is correct, it looks to me that is contradictory: (Line 264-265) ..."Conversely, about two thirds of HI and UMI countries and less than two thirds of UMI countries reported mesothelioma deaths, ..."
Our response: We apologize for this error. We corrected the sentence (Line 265-267): “Conversely, about two thirds of HI (68%) and UMI (65%) countries reported mesothelioma deaths, whereas the majority (86%) of LMI countries did not.“
Comment 3. If disease burden estimates for lower income countries have a larger effect on the global burden due to their population size, why not consider to extrapolate the number of inhabitants per country or its growth population rate to estimate of the country-level mesothelioma death burden?
What we meant by the statement “In general, disease burden estimates for lower income countries, relative to those for high-income countries, have a larger effect on the global burden due to their population size” is as follows: UMI and LMI generally have larger populations with China [1.39 Billion] (UMI) and India [1.35 Billion] (LMI) at the forefront. Consequently UMI and LMI (i.e., lower-income) countries will generally shoulder a larger burden of disease, that is the number of patients or deaths (for mesothelioma and any disease), relative to HI countries simply due to the larger population size of lower-income countries.
Just for clarification, in our previous work (Odgerel et al, 2017), to estimate the country-level mesothelioma death number, we extrapolated the gender- and age-specific mortality rates of the reporting countries with reliable data to the population size of the combined group of non-reporting countries by group-matching the level of cumulative and per capita asbestos consumption. This methodology prevents the size of population to become a source of bias.

Round 2
Reviewer 2 Report
explanation accepted